# *PPARgamma* Modulates CD4^+^ T-Cell Differentiation and Allergic Inflammation in Allergic Rhinitis: A Potential Therapeutic Target

**DOI:** 10.3390/biomedicines13071616

**Published:** 2025-07-01

**Authors:** Xiaoqing Rui, Suyu Ruan, Yu Zhang, Ranran Fu, Pengfei Sun, Danzeng Lamu, Weihua Wang

**Affiliations:** Department of Otolaryngology-Head and Neck Surgery, Shanghai East Hospital, School of Medicine, Tongji University, Shanghai 200092, China; ruixiaoqing@tongji.edu.cn (X.R.); ruansuyu310@163.com (S.R.); faust009@163.com (Y.Z.); 2233254@tongji.edu.cn (R.F.); 2233253@tongji.edu.cn (P.S.); 2332557@tongji.edu.cn (D.L.)

**Keywords:** allergic rhinitis, *PPARgamma*, naïve CD4^+^ T cells, regulatory T cells, T helper cells

## Abstract

**Objectives:** Given the emerging role of peroxisome proliferator-activated receptor gamma (*PPARgamma*) in immune regulation and the increasing prevalence of allergic rhinitis (AR), we sought to understand how modulation of the *PPARgamma* pathway impacts the balance of CD4^+^ T-cell subsets, particularly regulatory T cells (Tregs) and T helper (TH)1, TH2, and TH17 cells, which are key players in the pathogenesis of AR. This knowledge is crucial for developing novel therapeutic strategies targeting the *PPARgamma*-CD4^+^ T-cell axis to manage AR more effectively. **Methods:** We used *PPARgamma*^f/f^Lyz2-Cre mice for *PPARgamma* deletion. In an ovalbumin (OVA)-induced AR mouse model, *PPARgamma^+/-f/f^Lyz2*-Cre mice were assessed for allergic symptoms, splenic Tregs, and nasal eosinophils. Additionally, the effects of a *PPARgamma* agonist on the polarization of naïve CD4^+^ T cells were examined. **Results:** *PPARgamma^+/-f/f^Lyz2*-Cre mice showed worsened allergic symptoms, reduced splenic Tregs, and increased nasal mucosa eosinophilic infiltration. *PPARgamma* agonist treatment promoted naïve CD4^+^ T-cell polarization into Tregs and inhibited their differentiation into TH1, TH2, and TH17 subsets. **Conclusions:** Our findings indicate that *PPARgamma* plays a crucial role in regulating TH-cell subsets in AR. *PPARgamma* agonists could be a potential therapeutic strategy to mitigate allergic inflammation in AR by promoting Treg development and suppressing pathogenic TH-cell responses.

## 1. Introduction

Allergic rhinitis (AR) is a prevalent chronic nasal condition characterized by symptoms such as paroxysmal sneezing, rhinorrhea, nasal congestion, and nasal pruritus [1]. Its global prevalence ranges from 10% to 40% of the population [2]. Current treatment approaches for AR emphasize a “four in one” strategy that includes environmental control, pharmacotherapy, immunotherapy, and health education [3]. However, existing therapeutic options often fail to fully alleviate patients’ clinical symptoms, may have associated side effects, and are characterized by a high recurrence rate [4]. Therefore, there is an urgent need to explore safer and more effective treatment methods for AR.

Peroxisome proliferator-activated receptor gamma (*PPARgamma*) is a receptor transcription factor belonging to the nuclear hormone receptor superfamily [5]. The biological functions of *PPARgamma* are multifaceted, primarily involving the regulation of lipid and glucose metabolism [6]. Additionally, *PPARgamma* has a significant role in the regulation of inflammatory responses and immune modulation [7,8]. Studies have proven that *PPARgamma* is widely present in immune cells and can regulate inflammation [9]. It can influence immune cell differentiation and secretion, inhibit the expression of pro-inflammatory cytokines in macrophages, and modulate the type 1/type 2 immunity balance, thus playing an important role in inflammation and immune regulation [10,11]. In our previous study, we observed that the *PPARgamma* agonist significantly alleviated nasal allergy symptoms, reduced levels of interleukin (IL)-4, IL-5, and IL-13 in nasal lavage fluid, and inhibited the infiltration of inflammatory cells in the nasal mucosa [12].

The activation of naïve CD4^+^ T cells marks the beginning of their differentiation into mature, functionally specialized T helper (TH) cell subsets [13]. To date, several TH-cell subsets have been identified, including TH1, TH2, TH17, and regulatory T cells (Tregs). The balance between type 1 and type 2 immunity is crucial in the pathogenesis of allergic inflammation [14]. In parallel, extensive research has established that Tregs and associated cytokines play a fundamental role in maintaining immune homeostasis in allergic diseases [15,16]. Studies indicate that Tregs in cord blood exhibit impairments in quantity, expression, and function at birth, even before the development of type 1/type 2 immunity imbalances in the offspring of atopic mothers [17]. Furthermore, Tregs interact with TH17 cells to mediate allergic inflammation, as evidenced by their co-regulation in pediatric asthma and rhinitis. Prenatal exposure and genetic susceptibility contribute to early TH17 immune maturation, influencing childhood asthma development [18].

Despite evidence linking *PPARgamma* to T-cell regulation, two critical questions remain unaddressed: (i) Does myeloid *PPARgamma* deficiency exacerbate allergic inflammation by altering TH-subset dynamics? (ii) Can *PPARgamma* activation therapeutically reprogram CD4^+^ T-cell fate in AR? To investigate this, we first assessed symptoms and immune responses in ovalbumin (OVA)-induced AR using *PPARgamma* conditional knockout (cKO) mice (*PPARgamma^+/-f/f^Lyz2*-Cre) versus wild-type controls (*PPARgamma^-/-f/f^Lyz2*-Cre, wild type [WT]). We then evaluated splenic Treg proportions and nasal eosinophilia, and finally examined how *PPARgamma* agonism modulates naïve CD4^+^ T-cell polarization into Tregs, TH1, TH2, and TH17 lineages.

## 2. Materials and Methods

### 2.1. Mice

A total of 24 male C57BL/6 mice (approximately 8 weeks old) were utilized: 9 mice for in vivo studies comprising WT controls (*n* = 3), OVA-sensitized WT (*n* = 3), and myeloid-specific *PPARgamma* knockout mice (*n* = 3), and 15 WT mice for in vitro naïve CD4^+^ T-cell isolation. *PPARgamma*^f/f^Lyz2-Cre mice were generated to achieve myeloid-specific *PPARgamma* deletion by crossbreeding *PPARgamma*-floxed mice (*PPARgamma*^f/f^; *Pparg*tm2Rev/J) with Lyz2-Cre transgenic mice expressing Cre recombinase under the lysozyme 2 promoter. Both parental strains (C57BL/6 background; Stock No: C001003) were obtained from Cyagen Biosciences (Suzhou, China). The mice were housed under the standard conditions at a temperature of 22–24 °C with 50–60% humidity and a 12 h dark–light cycle. They were provided unrestricted access to OVA-free food and water. The study was approved by the Animal Research Ethics Board of Shanghai East Hospital (No. 2024-002).

### 2.2. OVA Sensitization, Antigen Challenge

Sensitization and antigen challenge were performed in mice using a modified established protocol [12]. Under pathogen-free conditions, mice in the AR model group were sensitized by the intraperitoneal (i.p.) injection of 150 μg OVA (Sigma-Aldrich, St. Louis, MO, USA) emulsified in 6 mg aluminum hydroxide [Al(OH)_3_] (Sango Biotech, Shanghai, China) in 0.3 mL sterile saline on days 0, 7, and 14. Control mice received i.p. injections of 0.3 mL sterile saline containing 6 mg Al(OH)_3_ (without OVA) on identical days. For nasal challenge, mice were administered 20 μL of 40 mg/mL OVA solution (800 μg/dose) via intranasal instillation (i.n.) daily from days 21 to 28. Control mice received 20 μL sterile saline using the same protocol (Figure 1). 

### 2.3. Symptom Assessment

Each mouse was challenged with an intranasal OVA instillation 24 h after the last intranasal saline or OVA challenge. Two observers blinded to experimental group assignments independently recorded the frequencies of sneezing and nasal rubbing events during the 15 min period immediately post-challenge. The final symptom counts represent the consensus value between observers, with discrepancies >15% resolved by video re-evaluation. This blinded protocol ensured the objective quantification of AR inflammatory parameters.

### 2.4. Measures of Tregs by Flow Cytometry

After OVA sensitization and allergen challenge, mice were deeply anesthetized with isoflurane overdose (5% induction, 3% maintenance) followed by cervical dislocation. Single-cell suspensions of spleens were prepared by passage through nylon mesh cell strainers (BD Biosciences, Franklin Lakes, NJ, USA). Single-cell suspensions from the spleen were prepared, and 1 × 10^6^ cells were used for flow cytometry. The cells were washed twice and resuspended in 100 μL of phosphate-buffered saline (PBS) and then stained with Percp/Cy5.5-conjugated anti-CD4 antibodies (Biolengend, San Diego, CA, USA) and incubated at 4 °C for 30 min. The resuspended cells were fixed and permeabilized using a Foxp3 Fix/Perm Fixation/Permeabilization Kit (Biolengend) under the manufacturer’s instructions. The cells were then stained with FITC-conjugated anti-Foxp3 antibodies (Biolengend) and incubated at 4 °C for 30 min. Flow cytometric analysis was performed using CytoFLEX and CytExpert software v2.4 (Beckman Coulter, Miami, FL, USA). The fluorescence strength was represented as a percentage. FoxP3^+^ CD4^+^ T cells were defined as Tregs.

### 2.5. Nasal Mucosa Histopathology Analysis

Following euthanasia, spleens were immediately harvested for flow cytometric analysis. Systemic perfusion fixation was then conducted via transcardial perfusion: approximately 40 mL of physiological saline followed by 20 mL of 4% paraformaldehyde (PFA) were administered through the left ventricle after puncturing the right auricle. Mouse heads were subsequently collected, trimmed of extraneous tissue, and post-fixed in 10% neutral-buffered formalin at 20 °C for 48 h. Fixed specimens underwent decalcification in 0.5 M EDTA solution for 28 days, with weekly solution changes and periodic tissue trimming to maintain optimal morphology. Following complete decalcification, specimens were paraffin-embedded. Serial 3 μm sections were stained with hematoxylin and eosin (H&E) to assess eosinophilic infiltration. Eosinophil quantification was performed by counting cells per high-power field (HPF) across three representative sections per mouse, with final values expressed as mean eosinophils/HPF.

### 2.6. Isolation of Naïve CD4^+^ T Cells and Measure of Treg/Th1/Th2/Th17 Cells by Flow Cytometric Analysis

Single-cell suspensions prepared from the spleens of untreated WT mice (genotype: *PPARgamma*^-/-f/f^Lyz2-Cre) were subjected to magnetic separation using a Naïve CD4^+^ T Cell Isolation kit (Miltenyi Biotech, Bergisch Gladbach, Germany; cat# 130-104-453), achieving >90% purity post-sorting; 6 × 10^6^ naïve CD4^+^ T cells were polarized for 72 h in anti-CD3ε-coated 96-well plates (5 μg/mL overnight) with soluble anti-CD28 (5 μg/mL) in 200 μL complete RPMI-1640 (10% FBS, 1 mM sodium pyruvate, 55 μM β-mercaptoethanol) under TH1 conditions (IL-2 [20 ng/mL] + IL-12 [20 ng/mL] + anti-IL-4 [10 μg/mL]); TH2 condition (IL-2 [20 ng/mL] + IL-4 [100 ng/mL] + anti-IFN-γ [10 μg/mL] + anti-IL-12 [10 μg/mL]); TH17 conditions (IL-2 [20 ng/mL] + IL-6 [100 ng/mL] + TGF-β [5 ng/mL] + anti-IL-4 [10 μg/mL] + anti-IFN-γ [10 μg/mL]), or Treg conditions (IL-2 [20 ng/mL] + TGF-β [5 ng/mL]), with pioglitazone (PIO) added at culture initiation and maintained throughout the 72 h polarization period. Cells were washed twice post-culture, surface-stained with anti-CD4 (BioLegend) at 4 °C for 30 min, fixed/permeabilized (Cytofix/Cytoperm Buffer Set, BD), intracellularly stained (BioLegend), and analyzed using CytoFLEX (CytExpert software v2.4); proportions were calculated as follows: TH1 = IFN-γ^+^CD4^+^, TH2 = IL-4^+^CD4^+^, TH17 = IL-17^+^CD4^+^, Tregs = FoxP3^+^CD4^+^.

### 2.7. Statistical Analysis

All quantitative data are presented as the mean ± standard deviation (SD). Statistical analyses were rigorously selected based on experimental design: one-way analysis of variance (ANOVA) with Tukey’s multiple comparisons test was employed for multi-group comparisons of independent samples, and Paired *t*-tests were used for two-group comparisons of related samples. Normality and homogeneity of variance assumptions were verified using Shapiro–Wilk and Brown–Forsythe tests, respectively. A probability value of *p* < 0.05 was considered statistically significant for all analyses, which were performed using GraphPad Prism 9.0 (GraphPad Software, San Diego, CA, USA).

## 3. Results

### 3.1. Myeloid PPARgamma Deficiency Exacerbates AR Pathology and Lowers Splenic Treg Proportion

Following OVA-induced AR modeling, the sneezing frequencies (15 min observation) were 6.00 ± 1.00 (WT Control), 37.33 ± 11.72 (WT AR), and 100.33 ± 10.97 (cKO-AR), while the nasal rubbing frequencies were 18.33 ± 6.66 (WT Control), 61.00 ± 17.56 (WT AR), and 83.33 ± 3.51 (cKO-AR). Statistical analysis using one-way ANOVA with Tukey’s post hoc test revealed significant group differences for sneezing (F = 80.31, *p* < 0.0001) and nasal rubbing (F = 79.38, *p* < 0.0001), with post hoc comparisons confirming increased sneezing in WT AR versus the control (*p* = 0.0145), cKO-AR versus the control (*p* < 0.0001), and cKO-AR versus WT AR (*p* = 0.0004), while nasal rubbing was elevated in both WT AR and cKO-AR versus the control (*p* < 0.001), and significantly higher in cKO-AR versus WT AR (*p* = 0.0126). Histopathological analysis demonstrated a disordered ciliated epithelium and significantly increased eosinophil infiltration in the nasal mucosa of cKO-AR mice (5.50 ± 0.89 eosinophils/HPF) versus WT AR (2.70 ± 0.78; *p* = 0.0102) and the control (0.63 ± 0.61; *p* < 0.0006) using ANOVA (F = 30.28, *p* = 0007) with Turkey’s multiple comparisons test. Flow cytometry confirmed reduced splenic Treg proportions in cKO-AR mice (6.16 ± 0.85%) versus WT AR (9.52 ± 1.47%; *p* = 0.0335) and the control (14.53 ± 1.23%; *p* = 0004), with overall significance indicated by ANOVA (F = 36.26, *p* = 0.0004) (Figure 2).

### 3.2. PPARgamma Agonist Promotes the Polarization of Naïve CD4^+^ T Cells into Tregs

Mouse spleen naïve CD4^+^ T cells were isolated via magnetic separation, achieving >90% purity (Figure 3A), and subsequently stimulated with CD3/CD28 in the presence of PIO (0, 1, 10, 20 μM) or a 0.1% DMSO vehicle. Flow cytometry analysis demonstrated concentration-dependent enhancement of Treg polarization (CD4^+^Foxp3^+^ cells), with significant overall concentration effects indicated by one-way ANOVA (F = 14.34, *p* = 0.0004), where Treg proportions at 1 μM (40.38 ± 6.28%; *p* = 0.019 vs. 0 μM), 10 μM (50.46 ± 3.04%; *p* < 0.0001 vs. 0 μM), and 20 μM (42.40 ± 5.94%; *p* = 0.003 vs. 0 μM) were all significantly elevated versus the 0 μM control (25.72 ± 0.67%) according to Tukey’s post hoc test, confirming maximal efficacy at 10 μM. This optimal effect was validated in expanded replicates, where a Paired *t*-test confirmed substantially higher Treg proportions at 10 μM versus the vehicle control (40.15 ± 4.13% vs. 28.20 ± 2.85%; t = 5.47, *p* = 0.0028) (Figure 3B,D).

### 3.3. PPARgamma Agonist Inhibits the Polarization of Naïve CD4^+^ T Cells Toward TH1/TH2/TH17 Cells

Naïve CD4^+^ T lymphocytes isolated from mouse spleens were stimulated with CD3/CD28 and polarized for 72 h under TH1, TH2, and TH17 conditions, with parallel treatment using 10 µM PIO or the DMSO vehicle. Paired *t*-test analysis revealed significantly reduced proportions of IFN-γ^+^ CD4^+^ T cells (TH1) (22.54 ± 2.41% vs. 28.12 ± 0.90%; t = 6.04, *p* = 0.0264) (Figure 4A,B) and IL-4^+^ CD4^+^ T cells (TH2) (2.73 ± 1.27% vs. 5.90 ± 1.33%; t = 14.83, *p* = 0.0045) (Figure 5A,B) in PIO-treated cultures versus the controls. While IL-17^+^ CD4^+^ T cells (TH17) showed no statistical significance (8.59 ± 4.03% vs. 12.16 ± 2.92%; t = 3.18, *p* = 0.0864), all three biological replicates demonstrated consistent downward trends (mean reduction: 3.57%) (Figure 6A,B).

## 4. Discussion

*PPARgamma*, a nuclear receptor superfamily member, has been studied in many inflammatory contexts [19,20]. Recent research shows that *PPARgamma* can suppress pro-inflammatory genes such as NF—κB [21] and MUC5AC [22,23,24]. *PPARgamma* agonists may also modulate inflammation by regulating miR expression; for example, *PPARgamma* upregulates miR—124 to inhibit pro-inflammatory cytokines [25]. Our experimental specifically addressed the functional role of myeloid *PPARgamma* in OVA-induced allergic inflammation through direct comparison of littermate-controlled *PPARgamma* cKO vs. WT mice under identical AR induction conditions. This design enables the definitive attribution of observed phenotypic differences—exacerbated symptoms, reduced Tregs, and enhanced eosinophil infiltration—to *PPARgamma* deficiency within the AR pathological context.

While our previous work demonstrated that systemic *PPARgamma* agonism enhances Tregs and ameliorates AR in murine models [12], and Park et al. reported that *PPARgamma* agonists inhibit TH1/TH2/TH17 differentiation with sex-dependent efficacy [26], these pharmacological approaches could not delineate the cellular origin of *PPARgamma*’s immunomodulatory effects. Our study resolves this gap through genetic dissection. This directly demonstrates that *PPARgamma* within myeloid lineages orchestrates T-cell polarization independently of systemic effects. Thus, our mechanistic insight advances prior pharmacological observations by identifying myeloid cells as the primary therapeutic target for *PPARgamma*-based interventions in AR.

We acknowledge that baseline characterization of *PPARgamma* cKO phenotypes was not included in the current study. However, this absence does not compromise our central finding that myeloid-specific *PPARgamma* deficiency exacerbates AR pathology, as all pathological and immunological endpoints were rigorously compared against genotype-matched, OVA-sensitized controls (cKO-AR vs. WT AR). Our experimental design specifically interrogated *PPARgamma*’s role within established allergic inflammation—not homeostatic immunity—making baseline cKO characterization non-essential for addressing the primary research question. Future studies examining steady-state immune profiles in unchallenged cKO mice would provide valuable mechanistic context, though such characterization extends beyond this work’s focus on *PPARgamma*-directed immunomodulation during active allergic responses.

The TH17-*PPARgamma* axis critically influences AR pathogenesis, where IL-17A elevation directly correlates with clinical severity in seasonal AR patients [27]. Our data align with *PPARgamma*’s immunomodulatory role: while PIO significantly suppressed TH1/TH2 polarization, it induced a consistent downward trend in TH17 proportions despite statistical non-significance, reinforcing *PPARgamma*’s capacity to broadly temper pro-inflammatory T-cell responses. Mechanistically, *PPARgamma* activation counteracts IL-17A-driven inflammation—amplified by pollutant-induced myeloid epigenetic dysregulation [28] and TLR4/NF-κB signaling [29,30]—while promoting Treg expansion. Therapeutically, *PPARgamma* agonism may rebalance TH17/Treg dynamics, addressing IL-17A-mediated neutrophil/eosinophil recruitment [31] unmitigated by current biologics.

Although our study did not directly assess epigenetic modifications, the observed effects of *PPARgamma* agonism on naïve CD4^+^T-cell polarization—promoting Treg differentiation while suppressing TH1/TH2/TH17 subsets—may involve epigenetic reprogramming. This possibility is supported by evidence that environmental exposure (e.g., air toxics) remodel the nasal mucosal epigenome, dysregulating genes controlling myeloid immune cell function and TGF-β1 signaling, which, in turn, drive aberrant TH cell responses in AR [16,27]. Moreover, multi-omics studies in AR have demonstrated coordinated alterations in DNA methylation and gene expression patterns in leukocytes during disease progression [28]. Future research should integrate transcriptomic and epigenomic profiling to determine whether *PPARgamma* activation durably recalibrates T-cell homeostasis through mechanisms such as the stable epigenetic silencing of pro-inflammatory pathways or enhancing regulatory circuitries. Such insights could inspire novel interventions targeting metabolic–epigenetic crosstalk in allergic inflammation.

Several limitations should be considered when interpreting our results. Firstly, the modest sample size in animal groups constrains the statistical power of our findings, particularly given the biological variability inherent in immune responses, necessitating expanded cohorts in future work. Secondly, murine AR models may not fully recapitulate human disease heterogeneity; the validation of *PPARgamma* expression in clinical nasal mucosa specimens and functional testing on human peripheral blood mononuclear cell (PBMC)-derived CD4^+^ T cells from AR patients are warranted to improve translational relevance. Finally, while IL-17^+^ cells were quantified, downstream mediators (e.g., CXCL chemokines) and functional consequences—including neutrophil recruitment and epithelial remodeling—remain unexamined and merit investigation in future work.

## 5. Conclusions

Myeloid *PPARgamma* deficiency exacerbates AR pathology and reduces splenic Tregs, while *PPARgamma* agonism enhances Treg polarization and suppresses TH1/TH2/TH17 differentiation. These results establish *PPARgamma* as a key regulator of allergic inflammation, supporting its therapeutic targeting to rebalance T-cell responses, warranting further preclinical and clinical validation.

## Figures and Tables

**Figure 1 biomedicines-13-01616-f001:**
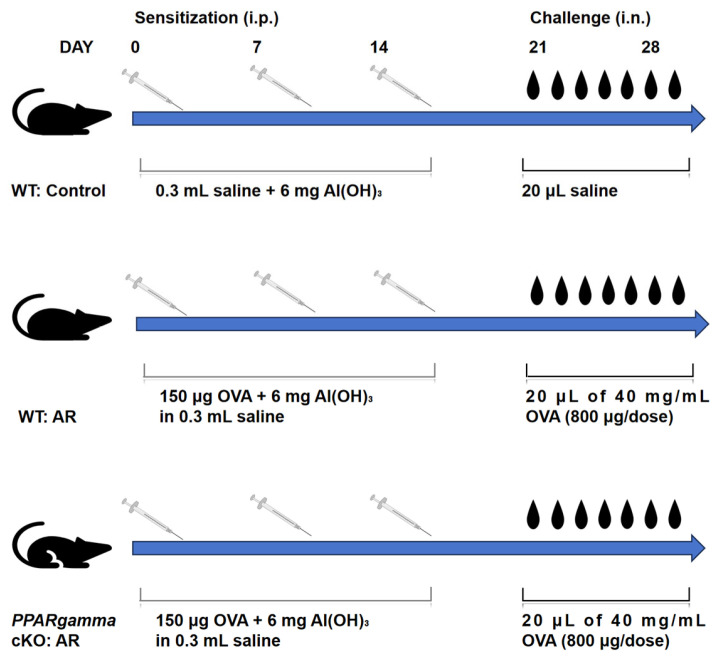
Schematic overview of in vivo experimental timeline for OVA-induced AR in myeloid-specific *PPARgamma*-deficient mice. Abbreviations: OVA, ovalbumin; i.p., intraperitoneal; i.n., intranasal; AR, allergic rhinitis; cKO, conditional knockout.

**Figure 2 biomedicines-13-01616-f002:**
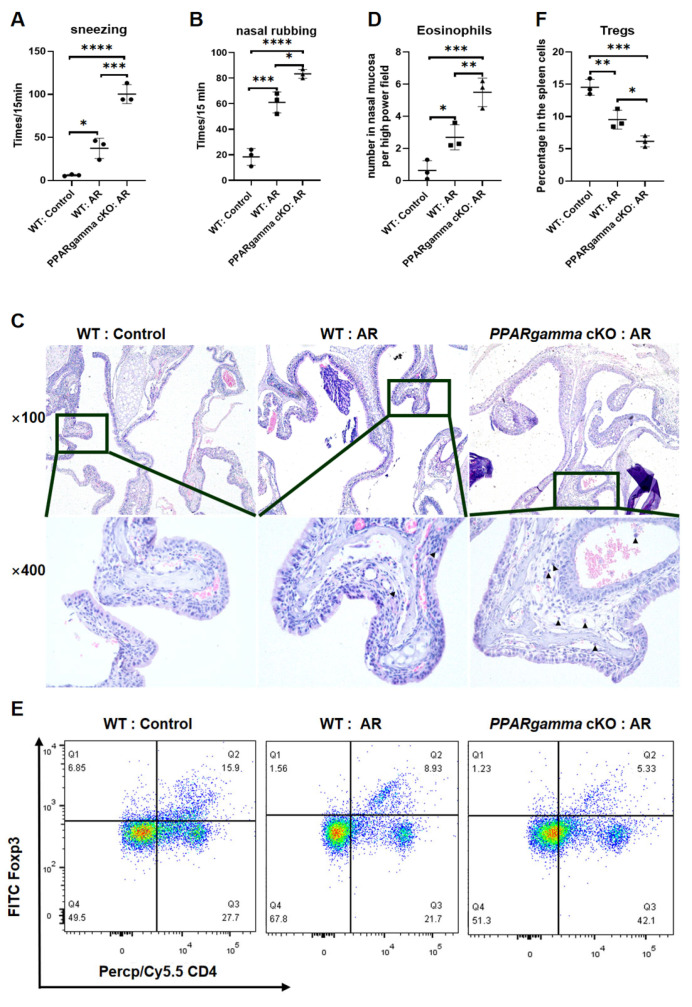
Myeloid *PPARgamma* deletion aggravates allergic responses in an OVA-induced AR model (*n* = 3 per group). (**A**,**B**) Allergic symptom frequencies in saline-treated controls, OVA-sensitized AR mice, and myeloid-specific *PPARgamma* cKO mice with OVA-induced AR (cKO-AR): (**A**) sneezing episodes; (**B**) nasal rubbing episodes. (**C**,**D**) Histopathological evaluation of nasal mucosa: (**C**) representative H&E-stained sections (100× and 400×; black arrows indicate eosinophils); (**D**) eosinophil quantification. Data represent mean eosinophil counts per mouse across three high-power fields (HPF, 400× magnification). (**E**,**F**) Splenic Treg analysis: (**E**) representative flow cytometry density plots (color gradient: red = high cell density, blue = low density) of Tregs (CD4^+^Foxp3^+^); (**F**) quantitative Treg proportion. Data represent mean ± SD. Statistical analysis: one-way ANOVA with Tukey’s post hoc test; * *p* < 0.05, ** *p* < 0.01, *** *p* < 0.001, **** *p* < 0.0001. Abbreviations: AR, allergic rhinitis; WT, wild type; OVA, ovalbumin; cKO, conditional knockout; H&E, hematoxylin and eosin; HPF, high-power field; SD, standard deviation.

**Figure 3 biomedicines-13-01616-f003:**
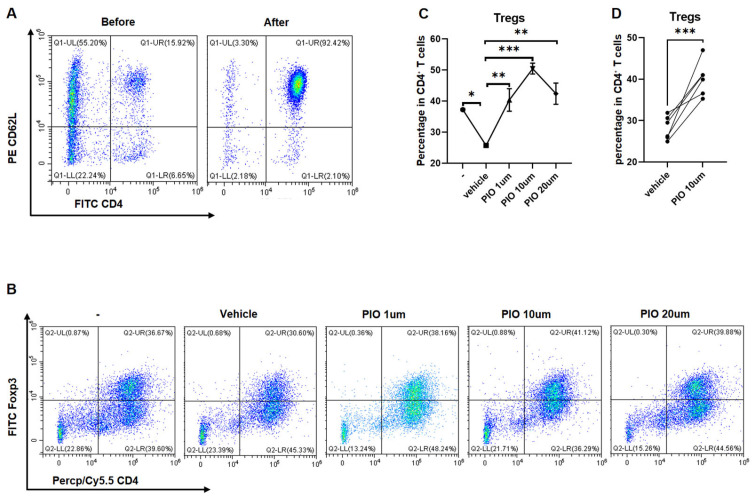
*PPARgamma* agonist enhances naïve CD4^+^ T-cell polarization toward Tregs. (**A**) Purity assessment of sorted naïve CD4^+^ T cells from mouse spleen. Representative flow cytometry density plots (color gradient: red = high cell density, blue = low density) show pre-sort (**left**) and post-sort (**right**) populations, with >90% purity achieved after magnetic-activated cell sorting (MACS). (**B**) Representative flow cytometry density plots of Treg polarization (CD4^+^Foxp3^+^ cells) under increasing PIO concentrations (0, 1, 10, 20 μM). (**C**) Concentration–response relationship quantifying Treg proportions from (**B**) (*n* = 3). (**D**) Validation of optimal concentration (10μM PIO) with expanded replication (*n* = 6). Data represent mean ± SD. Statistical analysis: one-way ANOVA with Tukey’s post hoc test (**C**) and Paired *t*-tests (**D**); * *p* < 0.05, ** *p* < 0.01, *** *p* < 0.001. Abbreviations: Tregs, regulatory T cells; PIO, pioglitazone; SD, standard deviation.

**Figure 4 biomedicines-13-01616-f004:**
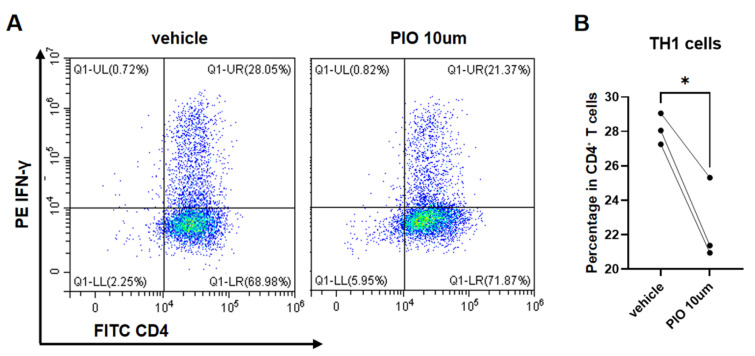
*PPARgamma* agonist suppresses TH1 polarization of naïve CD4^+^ T cells. (**A**) Representative flow cytometry density plots (color gradient: red = high cell density, blue = low density) of TH1 (CD4^+^IFN-γ^+^ cells) from naïve CD4^+^ T cells isolated from WT mice (*PPARgamma^-/-f/f^Lyz2*-Cre), cultured under TH1-polarizing conditions with PIO (10 μM) or DMSO vehicle control. (**B**) Quantitative analysis of TH1 proportions. Data represent mean ± SD. Statistical analysis: Paired *t*-test; * *p* < 0.05. Abbreviations: TH, T helper; PIO, pioglitazone; WT, wild type; SD, standard deviation.

**Figure 5 biomedicines-13-01616-f005:**
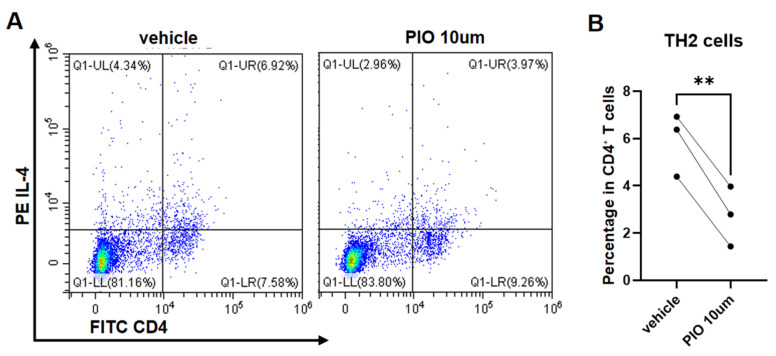
*PPARgamma* agonist suppresses TH2 polarization of naïve CD4^+^ T cells. (**A**) Representative flow cytometry density plots (color gradient: red = high cell density, blue = low density) of TH2 (CD4^+^IL-4^+^ cells) from naïve CD4^+^ T cells isolated from WT mice (*PPARgamma^-/-f/f^Lyz2*-Cre), cultured under TH2-polarizing conditions with PIO (10 μM) or DMSO vehicle control. (**B**) Quantitative analysis of TH2 proportions. Data represent mean ± SD. Statistical analysis: Paired *t*-test; ** *p* < 0.01. Abbreviations: TH, T helper; PIO, pioglitazone; WT, wild type; SD, standard deviation.

**Figure 6 biomedicines-13-01616-f006:**
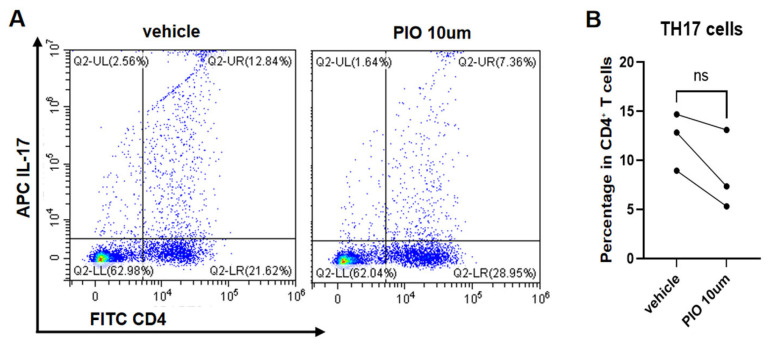
*PPARgamma* agonist suppresses TH17 polarization of naïve CD4^+^ T cells. (**A**) Representative flow cytometry density plots (color gradient: red = high cell density, blue = low density) of TH17 (CD4^+^IL-17^+^ cells) from naïve CD4^+^ T cells isolated from WT mice (*PPARgamma^-/-f/f^Lyz2*-Cre), cultured under TH17-polarizing conditions with PIO (10 μM) or DMSO vehicle control. (**B**) Quantitative analysis of TH17 proportions. Data represent mean ± SD. Statistical analysis: Paired *t*-test; ns: not significant. Abbreviations: TH, T helper; PIO, pioglitazone; WT, wild type; SD, standard deviation.

## Data Availability

The original contributions presented in this study are included in the article material. Further inquiries can be directed to the corresponding author.

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
