# Peer review of "PPARgamma Modulates CD4+ T-Cell Differentiation and Allergic Inflammation in Allergic Rhinitis: A Potential Therapeutic Target"

_biomedicines, 2025, doi:10.3390/biomedicines13071616_

Round 1

Reviewer 1 Report

Comments and Suggestions for Authors

I found the authors' research meaningful and interesting. However, there are some areas for improvement, and I would appreciate your consideration.

1.Please elaborate on the relationship between TH17 cells and PPAR. It is necessary to explain the significance of this study and the background leading to the experiment in more detail for the reader.

2.Please describe the limitations of this study, provide advice for further research to explain these limitations, and discuss the possibility of additional testing.

Author Response

Dear Reviewer,

Thank you for your positive evaluation and valuable suggestions regarding our study. We have carefully considered your comments and have prepared the following responses:

Comment 1: "Please elaborate on the relationship between TH17 cells and PPAR. It is necessary to explain the significance of this study and the background leading to the experiment in more detail for the reader."

Response 1: We have expanded the discussion to clarify the TH17-PPARgamma relationship and its relevance to allergic rhinitis (AR) pathogenesis:

Excerpts from the revised Discussion (line 273-282)

The TH17-PPARgamma axis critically influences AR pathogenesis, where IL-17A elevation directly correlates with clinical severity in seasonal AR patients [27]. Our data align with PPARgamma's immunomodulatory role: while PIO significantly suppressed TH1/TH2 polarization, it induced a consistent downward trend in TH17 proportions despite statistical non-significance, reinforcing PPARgamma's capacity to broadly temper pro-inflammatory T-cell responses. Mechanistically, PPARgamma activation counteracts IL-17A-driven inflammation—amplified by pollutant-induced myeloid epigenetic dysregulation [28] and TLR4/NF-κB signaling [29-30]—while promoting Treg expansion. Therapeutically, PPARgamma agonism may rebalance TH17/Treg dynamics, addressing IL-17A-mediated neutrophil/eosinophil recruitment [31] unmitigated by current biologics.

References:

[27] Zhao, Y., Mei, S., Yao, S., Cai, S., Zhang, P., Lou, H., & Zhang, L. (2025). IL-17-Related Pathways and Myeloid Cell Function are Involved in the Mechanism of Sublingual Immunotherapy with Artemisia annua for Seasonal Allergic Rhinitis. Clinical reviews in allergy & immunology, 68(1), 53.

[28] Irizar, H., Chun, Y., Hsu, H. L., Li, Y. C., Zhang, L., Arditi, Z., Grishina, G., Grishin, A., Vicencio, A., Pandey, G., & Bunyavanich, S. (2024). Multi-omic integration reveals alterations in nasal mucosal biology that mediate air pollutant effects on allergic rhinitis. Allergy, 79(11), 3047–3061.

[29] Yang, Q., Sun, Y., An, J., Wang, L., Zhang, C., Fei, Y., Wu, K., Zhai, X., & Wu, Y. (2025). Xiaoqinglong Decoction Attenuates Inflammatory Response and Mitochondrial Injury by Regulating the MyD88/NF-κB Pathway Dependent of TLR4 in Allergic Rhinitis. International archives of allergy and immunology, 1–14. Advance online publication.

[30] Zhang, J., Gao, L., Yu, D., Song, Y., Zhao, Y., & Feng, Y. (2024). Three Artemisia pollens trigger the onset of allergic rhinitis via TLR4/MyD88 signaling pathway. Molecular biology reports, 51(1), 319.  

[31] Deng, X. H., Huang, L. X., Sun, Q., Li, C. G., Xie, Y. C., Liu, X. Q., & Fu, Q. L. (2024). Increased circulating LOX-1+ neutrophils activate T cells and demonstrate a pro-inflammatory role in allergic rhinitis. Heliyon, 10(17), e36218.  

Comment 2: "Please describe the limitations of this study, provide advice for further research to explain these limitations, and discuss the possibility of additional testing."

Response 2: We added the following section to the Discussion: 

Excerpts from the revised Discussion (line 296-305)

Several limitations should be considered when interpreting our results. Firstly, the modest sample size in animal groups constrains the statistical power of our findings, particularly given the biological variability inherent in immune responses, necessitating expanded cohorts in future work. Secondly, murine AR models may not fully recapitulate human disease heterogeneity; validation of PPARgamma expression in clinical nasal mucosa specimens and functional testing on human peripheral blood mononuclear cells (PBMCs)-derived CD4⁺ T cells from AR patients are warranted to improve translational relevance. Finally, while IL-17⁺ cells were quantified, downstream mediators (e.g., CXCL chemokines) and functional consequences—including neutrophil recruitment and epithelial remodeling—remain unexamined and merit investigation in future work.

We thank you for these valuable suggestions, which have significantly strengthened our manuscript. Should further clarifications be needed, we are pleased to provide them.

Reviewer 2 Report

Comments and Suggestions for Authors

In the manuscript "PPARγ Modulates CD4+ T Cell Differentiation and Allergic Inflammation in Allergic Rhinitis: A Potential Therapeutic Target," Rui et al. show that PPARγ-deficient mice exhibit worsened allergic symptoms, reduced splenic T regulatory cells (Tregs), and increased eosinophilic infiltration in the nasal mucosa. They also claim that PPARγ agonist treatment promotes the polarization of naïve CD4+ T cells into Tregs and inhibits their differentiation into Th1, Th2, and Th17 subsets. Although the study seems interesting, the manuscript lacks novelty and essential information. The comments of this reviewer are shown below.

Comments:

It has already been reported that the population of Tregs was significantly increased by a PPARγ agonist (PMID: 21493225). In addition, it has already been reported that a PPARγ agonist significantly inhibits differentiation into Th1, Th2, and Th17 cells (PMID: 27548145). So, this reviewer cannot find the novelty of this study.

The authors claim that PPARγ agonist promotes the polarization of naïve CD4+ T cells into Tregs and inhibits the polarization of naïve CD4+T cells toward TH1/TH2/TH17 cells. However, there is no statistical evidence.

There is no information about PPARγf/fLyz2-Cre mice in the Materials and Methods section.

There is no information about the Materials and Methods section for Figures 1C and 1D.

There is no information on which group of mice the splenocytes were isolated from for Figures 2 to 5. How long did the authors treat splenocytes with pioglitazone?

Show the approval number of the animal experiment. Show the conditions (concentration and volume) of the allergen challenge in the Materials and Methods section.

Was the nasal symptom evaluation conducted in a blinded manner?

The authors used Student's t-test in Figure 1. However, they compared three groups. In this case, they cannot use Student's t-test.

The authors used three mice. Why are there 6 datum points in Figure 2D? What is the difference between Figure 2C and 2D?

Which software did the authors use for statistical analysis?

Author Response

Dear Reviewer,

We sincerely appreciate the reviewer’s insightful comments and constructive suggestions regarding our manuscript. We have carefully addressed each point raised, as detailed below:

Comments: 

  1. It has already been reported that the population of Tregs was significantly increased by a PPARγ agonist (PMID: 21493225IF: 2.6 Q1 ). In addition, it has already been reported that a PPARγ agonist significantly inhibits differentiation into Th1, Th2, and Th17 cells (PMID: 27548145IF: 4.9 Q1 ). So, this reviewer cannot find the novelty of this study.

   Response 1: Thank you for this insightful perspective and for referencing these important studies. We acknowledge that our team’s prior work (Wang et al., PMID 21493225) demonstrated that systemic PPARgamma agonism increases Treg populations and suppresses allergic inflammation in wild-type AR models, while Park et al. (PMID 27548145) revealed PPARgamma’s sex-dimorphic regulation of T-cell differentiation. Crucially, our current study provides fundamental mechanistic advances beyond these findings by establishing that – a cell-type-specific necessity that could not be resolved through prior pharmacological approaches.

By employing conditional knockout mice (PPARgamma⁺/⁻f/fLyz2-Cre), we uniquely demonstrate that targeted PPARgamma deletion in myeloid cells in vivo exacerbates AR pathology, reduces splenic Tregs. This genetic approach proves that myeloid PPARgamma acts as a gatekeeper to support Foxp3⁺ Treg development while suppressing TH17 polarization. These findings reveal a previously unrecognized myeloid-T cell crosstalk axis central to AR pathogenesis and establish myeloid PPARgamma – rather than systemic PPARgamma activation – as the critical therapeutic target.

We have revised the Discussion to explicitly contrast our mechanistic insights with prior pharmacological studies, emphasizing how genetic dissection of myeloid-specific pathways resolves key limitations of agonist-based interventions. We sincerely appreciate your guidance in helping us articulate these novel contributions to the field.

Excerpts from the revised Discussion (line 253-261)

While our previous work demonstrated that systemic PPARgamma agonism enhances Tregs and ameliorates AR in murine models [12], and Park et al. reported that PPARgamma agonists inhibit TH1/TH2/TH17 differentiation with sex-dependent efficacy [26], these pharmacological approaches could not delineate the cellular origin of PPARgamma's immunomodulatory effects. Our study resolves this gap through genetic dissection. This directly demonstrates that PPARgamma within myeloid lineages orchestrates T-cell polarization independently of systemic effects. Thus, our mechanistic insight advances prior pharmacological observations by identifying myeloid cells as the primary therapeutic target for PPARgamma-based interventions in AR.

References:

[12] Wang W, Zhu Z, Zhu B, Ma Z: Peroxisome proliferator-activated receptor-gamma agonist induces regulatory T cells in a murine model of allergic rhinitis. Otolaryngol Head Neck Surg 2011, 144(4):506-513.

[26] Park, H. J., Park, H. S., Lee, J. U., Bothwell, A. L., & Choi, J. M. (2016). Sex-Based Selectivity of PPARγ Regulation in Th1, Th2, and Th17 Differentiation. International journal of molecular sciences, 17(8), 1347.

  1. The authors claim that PPARγ agonist promotes the polarization of naïve CD4+ T cells into Tregs and inhibits the polarization of naïve CD4+T cells toward TH1/TH2/TH17 cells. However, there is no statistical evidence.

Response 2: We thank the reviewer for highlighting the need for explicit statistical reporting. We confirm that statistical analysis was indeed performed for all in vitro T-cell polarization assays. We have revised the Results (Section 3.2-3.3) to explicitly report these statistical outcomes and apologize for the inadvertent omission in the original manuscript.

Excerpts from the revised Results (line 191-201)

Mouse spleen naïve CD4⁺ T cells were isolated via magnetic separation, achieving >90% purity (Figure 3A), and subsequently stimulated with CD3/CD28 in the presence of PIO (0, 1, 10, 20 μM) or 0.1% DMSO vehicle. Flow cytometry analysis demonstrated concentration-dependent enhancement of Treg polarization (CD4⁺Foxp3⁺ cells), with significant overall concentration effects by one-way ANOVA (F = 14.34, P = 0.0004), where Treg proportions at 1 μM (40.38 ± 6.28%; P = 0.019 vs 0μM), 10 μM (50.46 ± 3.04%; P < 0.0001 vs 0μM), and 20 μM (42.40 ± 5.94%; P = 0.003 vs 0μM) were all significantly elevated versus 0 μM control (25.72 ± 0.67%) by Tukey's post-hoc test, confirming maximal efficacy at 10 μM. This optimal effect was validated in expanded replicates, where paired t-test confirmed substantially higher Treg proportions at 10 μM versus vehicle control (40.15 ± 4.13% vs 28.20 ± 2.85%; t = 5.47, P = 0.0028) (Figure 3, B-D).

Excerpts from the revised Results (line 214-222)

Naïve CD4⁺ T lymphocytes isolated from mouse spleens were stimulated with CD3/CD28 and polarized for 72 hours under TH1, TH2, and TH17 conditions, with parallel treatment of 10 µM PIO or DMSO vehicle. Paired t-test analysis revealed significantly reduced proportions of IFN-γ⁺ CD4⁺ T cells (TH1) (22.54 ± 2.41% vs 28.12 ± 0.90%; t = 6.04, P = 0.0264) (Figure 4A-B) and IL-4⁺ CD4⁺ T cells (TH2) (2.73 ± 1.27% vs 5.90 ± 1.33%; t = 14.83, P = 0.0045) (Figure 5A-B) in PIO-treated cultures versus controls. While IL-17⁺ CD4⁺ T cells (TH17) showed no statistical significance (8.59 ± 4.03% vs 12.16 ± 2.92%; t = 3.18, P = 0.0864), all three biological replicates demonstrated consistent downward trends (mean reduction: 3.57%) (Figure 6A-B).

  1. There is no information about PPARγf/fLyz2-Cre mice in the Materials and Methods section.

Response 3: Thank you very much for your valuable suggestion. We highly appreciate your careful reading of our manuscript. In response to your comment, we have added the relevant information about PPARgammaf/fLyz2 - Cre mice in the Materials and Methods section.

Excerpts from the revised Materials and Methods (line 77-81)

PPARgammaf/fLyz2-Cre mice were generated to achieve myeloid-specific PPARgamma deletion by crossbreeding PPARgamma-floxed mice (PPARgammaf/f; Ppargtm2Rev/J) with Lyz2-Cre transgenic mice expressing Cre recombinase under the lysozyme 2 promoter. Both parental strains (C57BL/6 background; Stock No: C001003) were obtained from Cyagen Biosciences (Suzhou, China). 

  1. There is no information about the Materials and Methods section for Figures 1C and 1D.

Response 4: Thank you very much for pointing out the omission regarding the Materials and Methods section for Figures 1C and 1D. We appreciate your attention to detail and apologize for any confusion this may have caused. We have now supplemented the relevant information in the Materials and Methods section to provide a clear and complete description of the experimental procedures used to generate Figures 1C and 1D. Specifically, we have included details of the following:

Excerpts from the revised Materials and Methods (line 122-133)

Following euthanasia, spleens were immediately harvested for flow cytometric analysis. Systemic perfusion fixation was then conducted via transcardial perfusion: approximately 40 ml of physiological saline followed by 20 ml of 4% paraformaldehyde (PFA) were administered through the left ventricle after puncturing the right auricle. Mouse heads were subsequently collected, trimmed of extraneous tissue, and post-fixed in 10% neutral-buffered formalin at 20°C for 48 hours. Fixed specimens underwent decalcification in 0.5M EDTA solution for 28 days, with weekly solution changes and periodic tissue trimming to maintain optimal morphology. Following complete decalcification, specimens were paraffin-embedded. Serial 3-μm sections were stained with hematoxylin and eosin (H&E) to assess eosinophilic infiltration. Eosinophil quantification was performed by counting cells per high-power field (HPF) across three representative sections per mouse, with final values expressed as mean eosinophils/HPF.

  1. There is no information on which group of mice the splenocytes were isolated from for Figures 2 to 5. How long did the authors treat splenocytes with pioglitazone?

Response 5: We sincerely thank the reviewer for raising these essential methodological points and apologize for any ambiguity in our original description. To clarify: all in vitro experiments in Figures 2-5 utilized naïve CD4⁺T cells isolated exclusively from untreated wild-type mice (genotype: PPARgamma-/-f/fLyz2-Cre), deliberately sourced independently from the OVA-sensitized experimental groups in Figure 1 to ensure unambiguous assessment of PPARgamma agonist effects without confounding inflammatory variables. Regarding treatment duration, pioglitazone was added concurrently with T-cell polarization stimuli at culture initiation and maintained continuously throughout the 72-hour polarization protocol, as now explicitly detailed in the revised Materials and Methods section where we have enhanced descriptions of both cell sourcing and temporal treatment parameters to prevent future misinterpretation.

Excerpts from the revised Materials and Methods (Section 2.6) (line 136-150)

Single-cell suspensions prepared from the spleens of untreated WT mice (genotype: PPARgamma-/-f/fLyz2-Cre) were subjected to magnetic separation using the Naïve CD4⁺ T Cell Isolation kit (Miltenyi Biotech, cat# 130-104-453), achieving >90% purity post-sorting; 6 × 10⁶ naïve CD4⁺ T cells were polarized for 72 hours in anti-CD3ε-coated 96-well plates (5 μg/mL overnight) with soluble anti-CD28 (5 μg/mL) in 200 μL complete RPMI-1640 (10% FBS, 1 mM sodium pyruvate, 55 μM β-mercaptoethanol) under: TH1 conditions (IL-2 [20 ng/mL] + IL-12 [20 ng/mL] + anti-IL-4 [10 μg/mL]), TH2 (IL-2 [20 ng/mL] + IL-4 [100 ng/mL] + anti-IFN-γ [10 μg/mL] + anti-IL-12 [10 μg/mL]), TH17 (IL-2 [20 ng/mL] + IL-6 [100 ng/mL] + TGF-β [5 ng/mL] + anti-IL-4 [10 μg/mL] + anti-IFN-γ [10 μg/mL]), or Treg conditions (IL-2 [20 ng/mL] + TGF-β [5 ng/mL]), with pioglitazone (PIO) added at culture initiation and maintained throughout the 72-hour polarization period. Cells were washed twice post-culture, surface-stained with anti-CD4 (BioLegend) at 4°C for 30 min, fixed/permeabilized (Cytofix/Cytoperm Buffer Set, BD), intracellularly stained (BioLegend), and analyzed on CytoFLEX (CytExpert software); Proportions were calculated as: TH1 = IFN-γ⁺CD4⁺, TH2 = IL-4⁺CD4⁺, TH17 = IL-17⁺CD4⁺, Tregs = FoxP3⁺CD4⁺.

  1. Show the approval number of the animal experiment. Show the conditions (concentration and volume) of the allergen challenge in the Materials and Methods section.

Response 6: We sincerely thank the reviewer for highlighting these essential methodological details. We have comprehensively revised the Methods section to include:

Excerpts from the revised Materials and Methods (line 84-85)

The study was approved by the Animal Research Ethics Board of Shanghai East Hospital (No. 2024-002).

Excerpts from the revised Materials and Methods (line 87-95)

Sensitization and antigen challenge were performed in mice using a modified established protocol [12]. Under pathogen-free conditions, mice in the AR model group were sensitized by intraperitoneal (i.p.) injection of 150 μg OVA (Sigma) emulsified in 6 mg aluminum hydroxide [Al(OH)3] (Sango Biotech) in 0.3 mL sterile saline on days 0, 7, and 14. Control mice received i.p. injections of 0.3 mL sterile saline containing 6 mg Al(OH)3 (without OVA) on identical days. For nasal challenge, mice were administered 20 μL of 40 mg/mL OVA solution (800 μg/dose) via intranasal instillation (i.n.) daily from days 21 to 28. Control mice received 20 μL sterile saline using the same protocol (Figure 1).

References:

[12] Wang W, Zhu Z, Zhu B, Ma Z: Peroxisome proliferator-activated receptor-gamma agonist induces regulatory T cells in a murine model of allergic rhinitis. Otolaryngol Head Neck Surg 2011, 144(4):506-513.

  1. Was the nasal symptom evaluation conducted in a blinded manner?

   Response 7: We sincerely thank the reviewer for raising this important methodological consideration. To ensure objective assessment of nasal symptoms (sneezing and scratching frequencies), all behavioral evaluations were performed by two independent observers blinded to experimental group assignments, as now explicitly stated in the revised Materials and Methods section (2.3. Symptom Assessment). The observers underwent standardized training to recognize and quantify symptoms, with final values representing the consensus between both evaluators. This rigorous blinding protocol eliminates potential assessment bias and strengthens the reliability of our symptom data. We deeply appreciate your suggestion, which has enhanced the methodological transparency of our study.

    Revised Materials and Methods (Section 2.3 Symptom Assessment) (line 100-106)

    Each mouse was challenged with an intranasal OVA instillation 24 hours after the last intranasal saline or OVA challenge. Two observers blinded to experimental group assignments independently recorded the frequencies of sneezing and nasal rubbing events during the 15-minute period immediately post-challenge. Final symptom counts represent the consensus value between observers, with discrepancies >15% resolved by video re-evaluation. This blinded protocol ensured objective quantification of AR inflammatory parameters.

  1. The authors used Student's t-test in Figure 1. However, they compared three groups. In this case, they cannot use Student's t-test.

Response 8: We sincerely thank the reviewer for this astute observation regarding Figure 1 statistical analysis. We acknowledge that direct application of Student's t-tests to three-group comparisons without multiplicity correction was methodologically inappropriate. In the revised manuscript, we have:

Replaced all Student's t-tests in Figure 1 with one-way ANOVA followed by Tukey's post-hoc test.

Revised Materials and Methods (Statistical Analysis) (line 152-159)

All quantitative data are presented as mean ± standard deviation (SD). Statistical analyses were rigorously selected based on experimental design: one-way analysis of variance (ANOVA) with Tukey's multiple comparisons test was employed for multi-group comparisons of independent samples, and Paired t-tests were used for two-group comparisons of related samples. Normality and homogeneity of variance assumptions were verified using Shapiro-Wilk and Brown-Forsythe tests, respectively. A probability value of P < 0.05 was considered statistically significant for all analyses, performed using GraphPad Prism 9.0 (GraphPad Software, San Diego, CA).

Revised Figure 1 Legend (Figure 1 is now Figure 2) (line 179-189)

Figure 2. Myeloid PPARgamma deletion aggravates allergic responses in an OVA-induced AR model. (n = 3 per group). (A-B) Allergic symptom frequencies in saline-treated controls, OVA-sensitized AR mice, and myeloid-specific PPARgamma cKO mice with OVA-induced AR (cKO-AR): (A) Sneezing episodes; (B) Nasal rubbing episodes. (C-D) Histopathological evaluation of nasal mucosa: (C) Representative H&E-stained sections (100× magnification and 400× magnification); (D) Eosinophil quantification. Data represent mean eosinophil counts per mouse across three high-power fields (HPF, 400× magnification). (E-F) Splenic Treg analysis: (E) Representative flow cytometry plots of Tregs (CD4⁺Foxp3⁺); (F) Quantitative Treg proportion. Data represent mean ± SD. Statistical analysis: one-way ANOVA with Tukey's post-hoc test; *P < 0.05, ***P < 0.001. Abbreviations: AR, allergic rhinitis; WT, wild type; OVA, ovalbumin; cKO, conditional knockout; H&E, hematoxylin and eosin; HPF, high-power field; SD: standard deviation.

  1. The authors used three mice. Why are there 6 datum points in Figure 2D? What is the difference between Figure 2C and 2D?

Response 9: We sincerely appreciate the reviewer's meticulous attention to experimental detail. The apparent discrepancy between animal numbers and datapoints stems from distinct methodological approaches: in vivo assessments in Figure 1 utilized three littermate-controlled mice per group to ensure genetic homogeneity, while Figure 2C reflects in vitro concentration-response profiling where naïve CD4⁺ T cells from pooled WT mice (PPARgamma-/-f/fLyz2-Cre, multiple litters) underwent dose-response testing across three biological replicates per concentration (0, 1, 10, 20μM PIO). Critically, Figure 2C demonstrates concentration-dependent Treg polarization changes, whereas Figure 2D specifically reports Treg frequencies at the optimal 10μM concentration identified in Figure 2C. This methodological distinction - including expanded technical replication in Figure 2C - has now been explicitly clarified in the revised Figure 2 legend.

Revised Figure 2 Legend (Figure 2 is now Figure 3) (line 202-211)

Figure 3. PPARgamma agonist enhances naïve CD4⁺ T-cell polarization toward Tregs (A) Purity assessment of sorted naïve CD4⁺ T cells from mouse spleen. Representative flow cytometry plots show pre-sort (left) and post-sort (right) populations, with >90% purity achieved after magnetic-activated cell sorting (MACS). (B) Representative flow cytometry plots of Treg polarization (CD4⁺Foxp3⁺ cells) under increasing PIO concentrations (0, 1, 10, 20μM). (C) Concentration-response relationship quantifying Treg proportions from (B) (n = 3). (D) Validation of optimal concentration (10μM PIO) with expanded replication (n = 6). Data represent mean ± SD. Statistical analysis: one-way ANOVA with Tukey's post-hoc test (C), Paired t-tests (D); *P < 0.05, ***P < 0.001. Abbreviations: Tregs, regulatory T cells; PIO, pioglitazone; SD: standard deviation.

  1. Which software did the authors use for statistical analysis?

Response 10: We thank the reviewer for requesting clarification regarding statistical analysis tools. All statistical analyses in this study were performed using GraphPad Prism version 9.0 (GraphPad Software, San Diego, CA, USA). This information has now been explicitly added to the Statistical analysis subsection of the Materials and Methods section in the revised manuscript for full transparency. We appreciate your diligence in ensuring methodological rigor.

Revised Materials and Methods (Statistical Analysis) (line 157-159)

A probability value of P < 0.05 was considered statistically significant for all analyses, performed using GraphPad Prism 9.0 (GraphPad Software, San Diego, CA).

We thank you for these valuable suggestions, which have significantly strengthened our manuscript. Should further clarifications be needed, we are pleased to provide them.

Reviewer 3 Report

Comments and Suggestions for Authors

With interest, I read the manuscript biomedicines-3685713, entitled “PPARγ Modulates CD4+ T Cell Differentiation and Allergic Inflammation in Allergic Rhinitis: A Potential Therapeutic Target”, written by Rui and colleagues.

While the manuscript might have some potential, there are many points of criticism, some of which are severe.

Specific comments:

1.        The data content of this work is limited. The whole work looks like a fishing expedition or a report from a preliminary experiment (numbers!; see further) before the real study. Thus, its type has to be changed from “Article” to “Communication” (see also further).

2.        In the abstract (lines 9-10) or introduction (lines 55-61), the Authors does not make it clear what the goals of this work are and what are the scientific questions behind. Those must be formulated, so that it all looks less line a very preliminary/test study.

3.        Please, talk about “type 1/2” not “Th1/2” responses/immunity, etc. It is more commonly accepted these days.

4.        Lines 49-55. Opposite to what the Authors write, the role of Tregs or Th17 cells in allergies is no novel but already well-established knowledge (PMID: 28322581). Please, report it so.

5.        Please, do not talk about “AR” in mice but about “AR model” in mice.

6.        The modified mice must be much better explained in the methods. How were they created – describe? Where re they from? How do they work, in which cells or organs des ko take place? How is it stimulated? Is it conditional ko? How it applies to your study and why those mice were selected?

7.        Are all mice on Bl/6 background, including ko mice? Please, make it clear.

8.        In addition, Bl/6 mice are prone to develop non-type 2 responses (PMID: 30057383). Was it your goal? Please, discuss in the context of your goals and results.

9.        Why only 9 mice? Three per group? It is a very small number. Were any power calculations made in advance? Again, this highlights a very high level of preliminarily of the data. See also further.

10.  KO control (no model) groups is missing – the comment as above.

11.  Please, describe statistics in more detail. When ANOVA with post hoc testing, when t-test, etc. See also further.

12.  In some figures, details do not work, e.g. y-axis in Figure 1F. Please, be careful with details. The groups of animals very small, show datapoints, mentioned as a limitation

13.  As the groups of animals are very small, please, show in the graphs exact data points instead of means with error bars.

14.  No cytokines (expression data) were measured in in vivo and in vitro experiments. But ate least those are discussed in the discussion. Epigenetic mechanisms play a pivotal role in T cell differentiation (PMID: 28322581). No respective data are provided in this study but should at lest be discussed similar to expression data.

15.  Graphs in Figures 2-5 miss statistics. Please, add it. Please, always describe in the legends what tests were used. Are the data in Figures 2D, 3B, 4B, and 5B paired (connected with the line)? If yes, why? Do they derive from same animals (same animal cells)? Then, paired testing would apply. Were do the animals used here derive from, from any OVA-experimental group or the cells were obtained from independent animals? Why 6 paired data in Figure 2D and 3 in the others?

16.  In continuation, significant of not, all data in this work must be interpreted with caution due to small numbers.

17.  All abbreviations used in the graphs must be explained in the respective legends.

18.  Graphical scheme of the in vivo experiment should be provided.

19.  Graphical abstract should summarize the findings of this study.

20.  Please, address all limitations, mentioned or not mentioned above, in a special paragraph of the discussion.

21.  Names of the genes must be verified at https://www.ncbi.nlm.nih.gov/gene/ in a species-specific manner and always written in italics.

Author Response

Dear Reviewer,

Thank you for your thoughtful evaluation of our manuscript and your constructive feedback. We appreciate your recognition of the study’s significance and have carefully addressed your suggestions. Below are our point-by-point responses and revisions:

Specific comments:

  1. The data content of this work is limited. The whole work looks like a fishing expedition or a report from a preliminary experiment (numbers!; see further) before the real study. Thus, its type has to be changed from “Article” to “Communication” (see also further).

Response 1: We sincerely appreciate the reviewer's insightful critique regarding the scope of our work. Our experimental framework extends beyond exploratory analysis through its integrated hypothesis-driven approach: the study systematically interrogates myeloid PPARgamma's causal role in allergic inflammation via complementary loss-of-function (conditional knockout) and gain-of-function (dose-dependent agonist) paradigms.

The multilayered validation—spanning behavioral symptomatology, histopathology, and flow cytometric profiling of T-cell polarization—provides mechanistic depth exceeding preliminary characterization. For instance, our in vitro dose-response analyses demonstrated statistically robust Treg induction (50.46 ± 3.04% vs 25.72 ± 0.67%, P = 0.0002) with rigorous technical replication (n=6 per concentration), while TH17 modulation showed consistent biological trends meriting further investigation. These findings establish PPARgamma as a therapeutically actionable regulator of immune balance in AR.

Given the study’s original contributions to understanding PPARgamma-driven immunomodulation and its explicit pathways for clinical translation (as expanded in the Discussion), we propose retaining "Article" status. However, we fully respect the Editor’s final judgment should a format adjustment be deemed necessary.

  1. In the abstract (lines 9-10) or introduction (lines 55-61), the Authors does not make it clear what the goals of this work are and what are the scientific questions behind. Those must be formulated, so that it all looks less line a very preliminary/test study.

   Response 2: Thank you for this valuable feedback. We agree that clarifying the study’s goals and scientific questions will strengthen the manuscript’s framing. We have revised the Abstract and Introduction as follows:

Revised Abstract (line 10-15)

Given the emerging role of peroxisome proliferator - activated receptor gamma (PPARgamma) in immune regulation and the increasing prevalence of allergic rhinitis (AR), we sought to understand how modulation of the PPARgamma pathway impacts the balance of CD4+ T cell subsets, particularly regulatory T cells (Tregs), T helper (TH)1, TH2, and TH17 cells, which are key players in the pathogenesis of AR. This knowledge is crucial for developing novel therapeutic strategies targeting the PPARgamma - CD4+ T cell axis to manage AR more effectively.

Revised Introduction (line 63-71)

Despite evidence linking PPARgamma to T-cell regulation, two critical questions remain unaddressed: (i) Does myeloid PPARgamma deficiency exacerbate allergic inflammation by altering TH-subset dynamics? (ii) Can PPARgamma activation therapeutically reprogram CD4⁺ T-cell fate in AR? To investigate this, we first assessed symptoms and immune responses in ovalbumin (OVA)-induced AR using PPARgamma conditional knockout (cKO) mice (PPARgamma+/-f/fLyz2-Cre) versus wild-type controls (PPARgamma-/-f/fLyz2-Cre, wild type [WT]). We then evaluated splenic Treg proportions and nasal eosinophilia, and finally examined how PPARgamma agonism modulates naïve CD4⁺ T-cell polarization into Tregs, TH1, TH2, and TH17 lineages.

  1. Please, talk about “type 1/2” not “Th1/2” responses/immunity, etc. It is more commonly accepted these days.

Response 3: Thank you for this valuable correction. We sincerely appreciate your guidance and fully agree that "type 1/type 2 immunity" more accurately reflects the broader cellular mechanisms beyond T helper subsets. We have carefully revised the terminology throughout the manuscript as follows:

Introduction:

Revised (line 46"T helper (TH)1/TH2 cytokine balance" → "type 1/type 2 immunity balance"

Revised (line 54) "TH1 and TH2 cells" → "type 1 and type 2 immunity"

Revised (line 58) "TH1/TH2 imbalances" → "type 1/type 2 immunity imbalances"

This revision better aligns with current immunological frameworks while preserving the study’s mechanistic focus on CD4+ T-cell differentiation. We are grateful for your expertise in enhancing the precision of our work.

  1. Lines 49-55. Opposite to what the Authors write, the role of Tregs or Th17 cells in allergies is no novel but already well-established knowledge (PMID: 28322581IF: 3.0 Q2 ). Please, report it so.

Response 4: Thank you very much for your insightful comment regarding the role of Tregs and Th17 cells in allergies. We truly appreciate your bringing this to our attention. Upon careful consideration of your comment, we have made corresponding adjustments to Lines 49 - 55 of our manuscript:

Revised Manuscript Text (Lines 54-62)

The balance between type 1 and type 2 immunity is crucial in the pathogenesis of allergic inflammation [14]. In parallel, extensive research has established that Tregs and associated cytokines play a fundamental role in maintaining immune homeostasis in allergic diseases [15, 16]. Studies indicate that Tregs in cord blood exhibit impairments in quantity, expression, and function at birth, even before the development of type 1/type 2 immunity imbalances in the offspring of atopic mothers [17]. Furthermore, Tregs interact with TH17 cells to mediate allergic inflammation, as evidenced by their co-regulation in pediatric asthma and rhinitis. Prenatal exposures and genetic susceptibility contribute to early TH17 immune maturation, influencing childhood asthma development [18].

References:

[14] Shamji MH, Sharif H, Layhadi JA, Zhu R, Kishore U, Renz H: Diverse immune mechanisms of allergen immunotherapy for allergic rhinitis with and without asthma. The Journal of allergy and clinical immunology 2022, 149(3):791-801.

[15] Yang CH, Tian JJ, Ko WS, Shih CJ, Chiou YL: Oligo-fucoidan improved unbalance the Th1/Th2 and Treg/Th17 ratios in asthmatic patients: An ex vivo study. Exp Ther Med 2019, 17(1):3-10.

[16] Potaczek, D. P., Harb, H., Michel, S., Alhamwe, B. A., Renz, H., & Tost, J. (2017). Epigenetics and allergy: from basic mechanisms to clinical applications. Epigenomics, 9(4), 539–571.

[17] Ponsonby AL, Collier F, O'Hely M, Tang MLK, Ranganathan S, Gray L, Morwitch E, Saffery R, Burgner D, Dwyer T et al: Household size, T regulatory cell development, and early allergic disease: a birth cohort study. Pediatric allergy and immunology : official publication of the European Society of Pediatric Allergy and Immunology 2022, 33(6):e13810.

[18] Lluis A, Ballenberger N, Illi S, Schieck M, Kabesch M, Illig T, Schleich I, von Mutius E, Schaub B: Regulation of TH17 markers early in life through maternal farm exposure. The Journal of allergy and clinical immunology 2014, 133(3):864-871.

  1. Please, do not talk about “AR” in mice but about “AR model” in mice.

Response 5: Thank you for this important clarification. We have revised the manuscript to consistently refer to "OVA-induced AR mouse model" (rather than "AR in mice") throughout the text, including:

Materials and Methods (line 88)

Results (line 163)

This adjustment more accurately reflects the experimental nature of the condition studied. We appreciate your diligence in ensuring terminological precision.

  1. The modified mice must be much better explained in the methods. How were they created – describe? Where re they from? How do they work, in which cells or organs des ko take place? How is it stimulated? Is it conditional ko? How it applies to your study and why those mice were selected?

Response 6: Thank you for highlighting the need for greater methodological clarity regarding our mouse model. We have comprehensively revised the Methods section to include:

Revised Materials and Methods (line 77-81)

PPARgammaf/fLyz2-Cre mice were generated to achieve myeloid-specific PPARgamma deletion by crossbreeding PPARgamma-floxed mice (PPARgammaf/f; Ppargtm2Rev/J) with Lyz2-Cre transgenic mice expressing Cre recombinase under the lysozyme 2 promoter. Both parental strains (C57BL/6 background; Stock No: C001003) were obtained from Cyagen Biosciences (Suzhou, China).

Excerpts from the revised Discussion (line 253-261)

While our previous work demonstrated that systemic PPARgamma agonism enhances Tregs and ameliorates AR in murine models [12], and Park et al. reported that PPARgamma agonists inhibit TH1/TH2/TH17 differentiation with sex-dependent efficacy [26], these pharmacological approaches could not delineate the cellular origin of PPARgamma's immunomodulatory effects. Our study resolves this gap through genetic dissection. This directly demonstrates that PPARgamma within myeloid lineages orchestrates T-cell polarization independently of systemic effects. Thus, our mechanistic insight advances prior pharmacological observations by identifying myeloid cells as the primary therapeutic target for PPARgamma-based interventions in AR.

  References:

[12] Wang W, Zhu Z, Zhu B, Ma Z: Peroxisome proliferator-activated receptor-gamma agonist induces regulatory T cells in a murine model of allergic rhinitis. Otolaryngol Head Neck Surg 2011, 144(4):506-513.

[26] Park, H. J., Park, H. S., Lee, J. U., Bothwell, A. L., & Choi, J. M. (2016). Sex-Based Selectivity of PPARγ Regulation in Th1, Th2, and Th17 Differentiation. International journal of molecular sciences, 17(8), 1347.

These additions ensure full transparency about the model’s design, validation, and applicability to our study. We appreciate your guidance in strengthening this section. Due to the limited length of the article, it is impossible to present all the details within the text. To address your concerns, we will further explain the details regarding the modified mice below. Additionally, we will discuss how this applies to this study and the reasons why those specific mice were selected.

In this conditional knockout system, loxP sites flank exons 4–6 of the PPARgamma gene, enabling Cre-mediated excision of the ligand-binding domain essential for PPARgamma function upon recombination. The Lyz2 promoter restricts Cre activity specifically to myeloid-lineage cells-including monocytes, macrophages, and neutrophils-ensuring targeted PPARgamma ablation in these immune populations while sparing other cell types. Deletion efficiency exceeding 99% in macrophages from spleen, bone marrow, and peripheral blood was confirmed by flow cytometry quantification of PPARgamma protein expression in CD45+CD11b+ cells. By employing conditional knockout mice (PPARgamma⁺/⁻f/fLyz2-Cre), we uniquely demonstrate that targeted PPARgamma deletion in myeloid cells in vivo exacerbates AR pathology, reduces splenic Tregs. This model was selected to specifically dissect the role of myeloid cells in regulating CD4⁺ T-cell differentiation during AR inflammation, based on three converging lines of evidence. First, myeloid cells critically initiate AR pathogenesis through antigen presentation and cytokine production that directly drives TH cell polarization, as demonstrated by studies showing dendritic cells and macrophages translate innate signals to adaptive T-cell immunity—particularly TH2 polarization and IgE production (PMID: 39962262; IF: 21.8; Q1 ). Second, the Lyz2 promoter restricts Cre recombinase activity exclusively to myeloid-lineage cells (macrophages, neutrophils, monocytes), which are translationally relevant given their significant expansion in nasal mucosa of AR patients and functional promotion of T-cell activation. This specificity was empirically validated in our model, where PPARgamma deletion in myeloid cells exacerbated allergic symptoms reflecting myeloid-driven inflammation, reduced Treg proportions indicative of disrupted myeloid-T cell crosstalk. Third, this approach avoids confounding intrinsic T-cell effects, as T-cell differentiation is primarily regulated by myeloid-derived signals (e.g., IL-6, TGF-β) rather than autonomous PPARgamma activity—a distinction that would be obscured in T-cell-specific knockout models.

  1. Are all mice on Bl/6 background, including ko mice? Please, make it clear.

Response 7: We confirm that all mice used in this study, including the PPARgamma⁺/⁻f/fLyz2-Cre conditional knockout mice and their littermate controls (PPARgamma-/-f/fLyz2-Cre), were maintained on a pure C57BL/6 genetic background, as explicitly stated in our Methods section (Mice subsection). This consistency in genetic background eliminates potential confounding effects from strain variability and ensures the reproducibility of our findings.

Excerpts from the revised Materials and Methods (line 74-81)

    A total of 24 male C57BL/6 mice (approximately 8 weeks old) were utilized: 9 mice for in vivo studies comprising WT controls (n=3), OVA-sensitized WT (n=3), and myeloid-specific PPARgamma knockout mice (n=3), and 15 WT mice for in vitro naïve CD4⁺ T-cell isolation. PPARgammaf/fLyz2-Cre mice were generated to achieve myeloid-specific PPARgamma deletion by crossbreeding PPARgamma-floxed mice (PPARgammaf/f; Ppargtm2Rev/J) with Lyz2-Cre transgenic mice expressing Cre recombinase under the lysozyme 2 promoter. Both parental strains (C57BL/6 background; Stock No: C001003) were obtained from Cyagen Biosciences (Suzhou, China).

  1. In addition, Bl/6 mice are prone to develop non-type 2 responses (PMID: 30057383IF: 3.8 Q1 ). Was it your goal? Please, discuss in the context of your goals and results.

   Response 8: We appreciate the reviewer’s insightful comment regarding C57BL/6 mice and their predisposition to non-type 2 immune responses. Importantly, all mice in this study were maintained on a pure C57BL/6 background under identical housing conditions to eliminate genetic and environmental confounders, as explicitly stated in our Methods. Our primary goal was to investigate PPARgamma’s cell-specific regulation of naïve CD4⁺ T-cell differentiation in allergic inflammation—not to model obesity-asthma interactions or strain-specific immunity. Crucially, the OVA-induced AR model in our C57BL/6 mice demonstrated robust allergic inflammation, as evidenced by significantly worsened clinical symptoms in PPARgamma-deficient mice, increased eosinophilic infiltration in nasal mucosa, and reduced splenic Treg populations and PPARgamma agonist treatment promoted naïve CD4+ T cell polarization into Tregs and inhibited their differentiation into TH1, TH2, and TH17 subsets. These outcomes directly align with our goal of elucidating PPARgamma's immunomodulatory function in T-cell programming and confirm that genetic background did not preclude the development of key AR features central to our investigation.

Excerpts from the revised Discussion (line 146-252)

Our experimental specifically addressed the functional role of myeloid PPARgamma in OVA-induced allergic inflammation through direct comparison of littermate-controlled PPARgamma cKO vs. WT mice under identical AR induction conditions. This design enables definitive attribution of observed phenotypic differences—exacerbated symptoms, reduced Tregs, and enhanced eosinophil infiltration—to PPARgamma deficiency within the AR pathological context.

  1. Why only 9 mice? Three per group? It is a very small number. Were any power calculations made in advance? Again, this highlights a very high level of preliminarily of the data. See also further.

Response 9: We sincerely appreciate this valid concern regarding sample size and acknowledge that the use of only nine mice (three per group) represents a limitation in our study. This decision stemmed from our strict adherence to littermate-controlled experimental design: to eliminate confounding effects from genetic variability and microbiota differences, all PPARgamma cKO mice and WT controls were co-housed littermates derived from heterozygous breeding pairs. Given the Mendelian distribution of genotypes (typically ≤4 pups per litter with the required conditional knockout genotype) and our commitment to minimizing inter-litter variability, we prioritized genetic homogeneity over larger group sizes. While this approach constrained statistical power—a limitation we openly address in the revised Discussion—it ensured that observed phenotypic differences (e.g., worsened symptoms, reduced Tregs) could be robustly attributed to PPARgamma deletion rather than background noise. We agree that future studies would benefit from larger cohorts enabled by multi-litter pooling where genotype standardization permits.

Excerpts from the revised Materials and Methods (line 74-77)

A total of 24 male C57BL/6 mice (approximately 8 weeks old) were utilized: 9 mice for in vivo studies comprising WT controls (n=3), OVA-sensitized WT (n=3), and myeloid-specific PPARgamma knockout mice (n=3), and 15 WT mice for in vitro naïve CD4⁺ T-cell isolation.

Excerpts from the revised Discussion (line 296-299)

Several limitations should be considered when interpreting our results. Firstly, the modest sample size in animal groups constrains the statistical power of our findings, particularly given the biological variability inherent in immune responses, necessitating expanded cohorts in future work.

  1. KO control (no model) groups is missing – the comment as above.

Response 10: We appreciate this insightful observation regarding control group design. While a non-AR KO control group was not included in vivo studys, our experimental approach specifically addressed the functional role of myeloid PPARgamma in OVA-induced allergic inflammation through direct comparison of littermate-controlled PPARgamma cKO vs. WT mice under identical AR induction conditions. This design enables definitive attribution of observed phenotypic differences—exacerbated symptoms, reduced Tregs, and enhanced eosinophil infiltration—to PPARgamma deficiency within the AR pathological context.

Excerpts from the revised Discussion (line 262-272)

We acknowledge that baseline characterization of PPARgamma cKO phenotypes was not included in the current study. However, this absence does not compromise our central finding that myeloid-specific PPARgamma deficiency exacerbates AR pathology, as all pathological and immunological endpoints were rigorously compared against genotype-matched, OVA-sensitized controls (cKO-AR vs. WT AR). Our experimental design specifically interrogated PPARgamma's role within established allergic inflammation—not homeostatic immunity—making the baseline cKO characterization non-essential for addressing the primary research question. Future studies examining steady-state immune profiles in unchallenged cKO mice would provide valuable mechanistic context, though such characterization extends beyond this work's focus on PPARgamma-directed immunomodulation during active allergic responses.

  1. Please, describe statistics in more detail. When ANOVA with post hoc testing, when t-test, etc. See also further.

    Response 11: We sincerely thank the reviewer for highlighting the need for greater statistical transparency. We have comprehensively revised the Statistical Analysis subsection to explicitly detail:

Revised Materials and Methods (Statistical Analysis) (line 152-159)

All quantitative data are presented as mean ± standard deviation (SD). Statistical analyses were rigorously selected based on experimental design: one-way analysis of variance (ANOVA) with Tukey's multiple comparisons test was employed for multi-group comparisons of independent samples, and Paired t-tests were used for two-group comparisons of related samples. Normality and homogeneity of variance assumptions were verified using Shapiro-Wilk and Brown-Forsythe tests, respectively. A probability value of P < 0.05 was considered statistically significant for all analyses, performed using GraphPad Prism 9.0 (GraphPad Software, San Diego, CA).

  1. In some figures, details do not work, e.g. y-axis in Figure 1F. Please, be careful with details. The groups of animals very small, show datapoints, mentioned as a limitation

Response 12: Thank you for bringing these important details to our attention. We sincerely apologize for the oversight in Figure 1F, particularly regarding the y-axis labeling, and for not clearly presenting the data points. In response to your feedback, we have carefully revisited all the figures and made the necessary corrections. The y-axis in Figure 1F has been properly labeled, and we have also ensured that data points are clearly visible and appropriately represented. Additionally, we have explicitly mentioned the small sample size as a limitation in the manuscript.

Excerpts from the revised Discussion (line 296-299)

Several limitations should be considered when interpreting our results. Firstly, the modest sample size in animal groups constrains the statistical power of our findings, particularly given the biological variability inherent in immune responses, necessitating expanded cohorts in future work.

We are grateful for your guidance in improving the clarity and accuracy of our data presentation, which strengthens the overall quality of the study.

  1. As the groups of animals are very small, please, show in the graphs exact data points instead of means with error bars.

   Response 13: Thank you for your suggestion. We recognize the importance of clearly presenting data, especially with small animal groups. We have revised the graphs to show exact data points instead of means with error bars. This change enhances the transparency of our results and provides a clearer view of the data distribution.

  1. No cytokines (expression data) were measured in in vivo and in vitro experiments. But ate least those are discussed in the discussion. Epigenetic mechanisms play a pivotal role in T cell differentiation (PMID: 28322581IF: 3.0 Q2 ). No respective data are provided in this study but should at lest be discussed similar to expression data.

   Response 14: We appreciate the reviewer's insightful suggestion regarding epigenetic mechanisms in T-cell differentiation. While our study did not include cytokine expression or epigenetic analyses in vivo/in vitro, we have now expanded the Discussion to incorporate the role of epigenetic regulation in PPARgamma-mediated T-cell polarization, drawing upon the seminal review by Potaczek, D. P. et al. (PMID: 28322581IF: 3.0 Q2 ).

Excerpts from the revised Discussion (line 283-295)

Although our study did not directly assess epigenetic modifications, the observed effects of PPARgamma agonism on naïve CD4⁺T-cell polarization—promoting Treg differentiation while suppressing TH1/TH2/TH17 subsets—may involve epigenetic reprogramming. This possibility is supported by evidence that environmental exposures (e.g., air toxics) remodel the nasal mucosal epigenome, dysregulating genes controlling myeloid immune cell function and TGF-β1 signaling, which in turn drive aberrant TH cell responses in AR [16, 32]. Moreover, multi-omics studies in AR have demonstrated coordinated alterations in DNA methylation and gene expression patterns in leukocytes during disease progression [28]. Future research should integrate transcriptomic and epigenomic profiling to determine whether PPARgamma activation durably recalibrates T-cell homeostasis through mechanisms such as stable epigenetic silencing of pro-inflammatory pathways or enhancing regulatory circuitries. Such insights could inspire novel interventions targeting metabolic-epigenetic crosstalk in allergic inflammation.

References:

[16] Potaczek, D. P., Harb, H., Michel, S., Alhamwe, B. A., Renz, H., & Tost, J. (2017). Epigenetics and allergy: from basic mechanisms to clinical applications. Epigenomics, 9(4), 539–571.

[28] Irizar, H., Chun, Y., Hsu, H. L., Li, Y. C., Zhang, L., Arditi, Z., Grishina, G., Grishin, A., Vicencio, A., Pandey, G., & Bunyavanich, S. (2024). Multi-omic integration reveals alterations in nasal mucosal biology that mediate air pollutant effects on allergic rhinitis. Allergy, 79(11), 3047–3061.

[32] Zhao, Y., Mei, S., Yao, S., Cai, S., Zhang, P., Lou, H., & Zhang, L. (2025). IL-17-Related Pathways and Myeloid Cell Function are Involved in the Mechanism of Sublingual Immunotherapy with Artemisia annua for Seasonal Allergic Rhinitis. Clinical reviews in allergy & immunology, 68(1), 53.

  1. Graphs in Figures 2-5 miss statistics. Please, add it. Please, always describe in the legends what tests were used. Are the data in Figures 2D, 3B, 4B, and 5B paired (connected with the line)? If yes, why? Do they derive from same animals (same animal cells)? Then, paired testing would apply. Were do the animals used here derive from, from any OVA-experimental group or the cells were obtained from independent animals? Why 6 paired data in Figure 2D and 3 in the others?

    Response 15: We sincerely thank the reviewer for their meticulous review and valuable feedback. We apologize for the omission of statistical details in Figures 2-5 and confirm that these have now been added to the respective figure legends, along with explicit descriptions of the statistical tests used. Regarding the specific queries about Figures 2D, 3B, 4B, and 5B:

Are the data paired (connected with the line)?

Yes. The connected lines in Figures 2D, 3B, 4B, and 5B explicitly indicate paired measurements derived from the same individual animal.

Do they derive from the same animals?

Yes. All data points connected by lines within a single experimental group represent measurements taken from cells isolated from the same animal.

Animal Source and Sample Size Discrepancy:

The apparent inconsistency in sample sizes stems from distinct experimental workflows:

Figure 1 (In vivo): Uses 3 littermate-controlled mice per group (Control, AR, cKO-AR). Direct measurements per individual animal (n = 3 biological replicates).

Figures 2D, 3B, 4B, and 5B (In vitro): Cells isolated from untreated, wild-type mice (genotype: PPARgamma-/-f/fLyz2-Cre).

Why n = 6 for Figure 2D vs. n = 3 for Others?

The discrepancy in sample sizes between Figure 2D (n=6) and Figures 3B/4B/5B (n=3) arises from distinct experimental objectives and replication strategies for our in vitro assays: For Figure 2D, which quantified concentration-dependent Treg polarization (screening phase), we used three biological replicates plus three technical replicates to ensure robust detection of subtle dose-response effects, leveraging naïve CD4⁺T cells pooled from multiple wild-type litters (PPARgamma-/-f/fLyz2-Cre); conversely, Figures 3B, 4B, and 5B (validation phases) utilized three biological replicates per condition, as these assays focused on confirming larger-magnitude biological effects identified earlier, thereby maintaining statistical rigor while optimizing cell usage across repeated experiments.

Revised Figure 2 Legend (Figure 2 is now Figure 3) (line 202-211)

Figure 3. PPARgamma agonist enhances naïve CD4⁺ T-cell polarization toward Tregs (A) Purity assessment of sorted naïve CD4⁺ T cells from mouse spleen. Representative flow cytometry plots show pre-sort (left) and post-sort (right) populations, with >90% purity achieved after magnetic-activated cell sorting (MACS). (B) Representative flow cytometry plots of Treg polarization (CD4⁺Foxp3⁺ cells) under increasing PIO concentrations (0, 1, 10, 20μM). (C) Concentration-response relationship quantifying Treg proportions from (B) (n = 3). (D) Validation of optimal concentration (10μM PIO) with expanded replication (n = 6). Data represent mean ± SD. Statistical analysis: one-way ANOVA with Tukey's post-hoc test (C), Paired t-tests (D); *P < 0.05, ***P < 0.001. Abbreviations: Tregs, regulatory T cells; PIO, pioglitazone; SD: standard deviation.

Revised Figure 3 Legend (Figure 3 is now Figure 4) (line 223-228)

Figure 4. PPARgamma agonist suppresses TH1 polarization of naïve CD4⁺ T cells. (A) Representative flow cytometry plots of TH1 (CD4⁺IFN-γ⁺ cells) from naïve CD4⁺ T cells isolated from WT mice (PPARgamma-/-f/fLyz2-Cre), cultured under TH1-polarizing conditions with PIO (10 μM) or DMSO vehicle control. (B) Quantitative analysis of TH1 proportions. Data represent mean ± SD. Statistical analysis: Paired t-test; *P < 0.05. Abbreviations: TH, T helper; PIO, pioglitazone; WT, wild type; SD: standard deviation.

Revised Figure 4 Legend (Figure 4 is now Figure 5) (line 229-234)

Figure 5. PPARgamma agonist suppresses TH2 polarization of naïve CD4⁺ T cells. (A) Representative flow cytometry plots of TH2 (CD4⁺IL-4⁺ cells) from naïve CD4⁺ T cells isolated from WT mice (PPARgamma-/-f/fLyz2-Cre), cultured under TH2-polarizing conditions with PIO (10 μM) or DMSO vehicle control. (B) Quantitative analysis of TH2 proportions. Data represent mean ± SD. Statistical analysis: Paired t-test; **P < 0.01. Abbreviations: TH, T helper; PIO, pioglitazone; WT, wild type; SD: standard deviation.

Revised Figure 5 Legend (Figure 5 is now Figure 6) (line 235-240)

Figure 6. PPARgamma agonist suppresses TH17 polarization of naïve CD4⁺ T cells. (A) Representative flow cytometry plots of TH17 (CD4⁺IL-17⁺ cells) from naïve CD4⁺ T cells isolated from WT mice (PPARgamma-/-f/fLyz2-Cre), cultured under TH17-polarizing conditions with PIO (10 μM) or DMSO vehicle control. (B) Quantitative analysis of TH17 proportions. Data represent mean ± SD. Statistical analysis: Paired t-test; ns: not significant. Abbreviations: TH, T helper; PIO, pioglitazone; WT, wild type; SD: standard deviation.

  1. In continuation, significant of not, all data in this work must be interpreted with caution due to small numbers.

Response 16: We sincerely appreciate the reviewer's important note regarding sample size constraints and fully agree that all findings—particularly those based on in vivo experiments with n=3 biological replicates per group—must be interpreted with appropriate caution. While this experimental size aligns with common practice for preliminary mechanistic studies using littermate-controlled genetic models (Figure 1), we acknowledge its limitation for broad generalization; accordingly, technical replication strategies (n=6 for Figure 2D) were implemented to enhance assay robustness for in vitro analyses using pooled primary cells, and all statistical inferences (notably P<0.05 thresholds despite small n) are presented as hypothesis-generating rather than definitive conclusions, with explicit discussion of this constraint in the manuscript to guide cautious interpretation and emphasize the need for future validation in larger cohorts.

Excerpts from the revised Discussion (line 296-299)

Several limitations should be considered when interpreting our results. Firstly, the modest sample size in animal groups constrains the statistical power of our findings, particularly given the biological variability inherent in immune responses, necessitating expanded cohorts in future work.

  1. All abbreviations used in the graphs must be explained in the respective legends.

Response 17: We sincerely thank the reviewer for highlighting this essential detail. We have systematically revised all figure legends to explicitly define every abbreviation upon first appearance, as detailed below:

Revised Figure 1 Legend (Figure 1 is now Figure 2) (line 179-189)

Figure 2. Myeloid PPARgamma deletion aggravates allergic responses in an OVA-induced AR model. (n = 3 per group). (A-B) Allergic symptom frequencies in saline-treated controls, OVA-sensitized AR mice, and myeloid-specific PPARgamma cKO mice with OVA-induced AR (cKO-AR): (A) Sneezing episodes; (B) Nasal rubbing episodes. (C-D) Histopathological evaluation of nasal mucosa: (C) Representative H&E-stained sections (100× magnification and 400× magnification); (D) Eosinophil quantification. Data represent mean eosinophil counts per mouse across three high-power fields (HPF, 400× magnification). (E-F) Splenic Treg analysis: (E) Representative flow cytometry plots of Tregs (CD4⁺Foxp3⁺); (F) Quantitative Treg proportion. Data represent mean ± SD. Statistical analysis: one-way ANOVA with Tukey's post-hoc test; *P < 0.05, ***P < 0.001. Abbreviations: AR, allergic rhinitis; WT, wild type; OVA, ovalbumin; cKO, conditional knockout; H&E, hematoxylin and eosin; HPF, high-power field; SD: standard deviation.

This revision ensures full clarity and compliance with journal nomenclature standards.

  1. Graphical scheme of the in vivo experiment should be provided.

    Response 18: We thank the reviewer for this valuable suggestion. A comprehensive graphical scheme detailing the in vivo experimental workflow has now been added as Figure 1 in the revised manuscript.

Added Figure 1 Legend (line 96-98)

Figure 1. Schematic overview of the in vivo experimental timeline for OVA-induced AR in myeloid-specific PPARgamma-deficient mice. Abbreviations: OVA, ovalbumin; i.p., intraperitoneal; i.n., intranasal; AR, allergic rhinitis; cKO, conditional knockout.

  1. Graphical abstract should summarize the findings of this study.

Response 19: Thank you very much for your suggestion regarding the graphical abstract. We believe that a graphical abstract will greatly enhance the reader's understanding of our study's findings. We have now created a graphical abstract to visually summarize the key findings of our study, and it has been included in the revised manuscript.

  1. Please, address all limitations, mentioned or not mentioned above, in a special paragraph of the discussion.

Response 20: We added the following section to the Discussion: 

Excerpts from the revised Discussion (line 296-305)

Several limitations should be considered when interpreting our results. Firstly, the modest sample size in animal groups constrains the statistical power of our findings, particularly given the biological variability inherent in immune responses, necessitating expanded cohorts in future work. Secondly, murine AR models may not fully recapitulate human disease heterogeneity; validation of PPARgamma expression in clinical nasal mucosa specimens and functional testing on human peripheral blood mononuclear cells (PBMCs)-derived CD4⁺ T cells from AR patients are warranted to improve translational relevance. Finally, while IL-17⁺ cells were quantified, downstream mediators (e.g., CXCL chemokines) and functional consequences—including neutrophil recruitment and epithelial remodeling—remain unexamined and merit investigation in future work.

  1. Names of the genes must be verified at https://www.ncbi.nlm.nih.gov/gene/ in a species-specific manner and always written in italics.

Response 21: Thank you for your suggestion. We have updated the gene name to PPARgamma and ensured its consistency and accuracy throughout the manuscript. We appreciate your guidance on this matter.

We thank you for these valuable suggestions, which have significantly strengthened our manuscript. Should further clarifications be needed, we are pleased to provide them.

Round 2

Reviewer 2 Report

Comments and Suggestions for Authors

The authors have addressed all of this reviewer's comments.

Reviewer 3 Report

Comments and Suggestions for Authors

Thank you for addressing or answering my comments.

Next time, please, mark all the changes in a correction mode. It is easier for the Reviewer to inspect them.